# Updated Gene Prediction of the Cucumber (9930) Genome through Manual Annotation

**DOI:** 10.3390/plants13121604

**Published:** 2024-06-09

**Authors:** Weixuan Du, Lei Xia, Rui Li, Xiaokun Zhao, Danna Jin, Xiaoning Wang, Yun Pei, Rong Zhou, Jinfeng Chen, Xiaqing Yu

**Affiliations:** 1State Key Laboratory of Crop Genetics & Germplasm Enhancement and Utilization, Nanjing Agricultural University, No. 1 Weigang, Nanjing 210095, Chinajfchen@njau.edu.cn (J.C.); 2College of Agriculture, Guizhou University, Guiyang 550025, China; 3Department of Food Science, Plant, Food & Climate, Aarhus University, Agro Food Park 48, DK-8200 Aarhus, Denmark; rong.zhou@food.au.dk

**Keywords:** cucumber, manual reannotation, transcriptome, gene model, tissue-specific expression

## Abstract

Thorough and precise gene structure annotations are essential for maximizing the benefits of genomic data and unveiling valuable genetic insights. The cucumber genome was first released in 2009 and updated in 2019. To increase the accuracy of the predicted gene models, 64 published RNA-seq data and 9 new strand-specific RNA-seq data from multiple tissues were used for manual comparison with the gene models. The updated annotation file (V3.1) contains an increased number (24,145) of predicted genes compared to the previous version (24,317 genes), with a higher BUSCO value of 96.9%. A total of 6231 and 1490 transcripts were adjusted and newly added, respectively, accounting for 31.99% of the overall gene tally. These newly added and adjusted genes were renamed (CsaV3.1_XGXXXXX), while genes remaining unaltered preserved their original designations. A random selection of 21 modified/added genes were validated using RT-PCR analyses. Additionally, tissue-specific patterns of gene expression were examined using the newly obtained transcriptome data with the revised gene prediction model. This improved annotation of the cucumber genome will provide essential and accurate resources for studies in cucumber.

## 1. Introduction

The utility of genomic data relies on precise and comprehensive genome annotation. This process involves multifaceted gene prediction, drawing from diverse data sources [1]. Common strategies for genome annotation encompass de novo prediction, homology prediction via cross-species protein sequences, and automated prediction utilizing RNA data from varied tissues [2,3]. Genomes of some important species have been subjected to repeated reannotation efforts to enrich genetic information exploration. Notably, recent years have witnessed multiple versions of genome annotations for Arabidopsis, citrus, rice, peach, and maize [4,5,6,7,8]. Manual refinement of gene models through RNA data visualization to repair shortages of algorithmic software has proven efficacious in enhancing the accuracy of annotation files. WebApollo software (version 2.0.2) has been employed to manually adjust gene models for the kiwifruit genome [9], and the GSAman software (v.0.8.2) has been utilized to refine gene structure models for the peach and sweet potato genomes [8,10].

Cucumber (*Cucumis sativus* L.) is a widely cultivated annual herbaceous plant recognized as one of the top ten globally grown vegetables due to its distinctive taste and nutritionally dense profile. Cucumber was the first vegetable crop to be successfully sequenced in 2009, marking the onset of the genomic era in vegetable research and facilitating molecular breeding advancements [11]. Through whole-genome shotgun sequencing techniques, including Sanger and next-generation sequencing, a comprehensive genome sequence of 243.5 Mb was assembled for the cucumber 9930 (‘Chinese Long’ line). This effort validated the evolutionary transformation of cucumber’s seven chromosomes from twelve ancestral chromosomes and laid the foundation for subsequent comparative genomics studies on the identification of trait-related genes that are favored during cucumber domestication [11,12]. Key genes linked to the generation of bitterness in cucumbers were pinpointed, effectively resolving the issue of bitter cucumbers in the southern region of China and producing notable societal advantages [13].

The release of the 9930_V3 version of the cucumber reference genome, generated using PacBio third-generation sequencing technology, represents a significant improvement in the research in cucumber-related fields. However, the current gene annotation for the 9930_V3 reference genome is based on algorithmic predictions, highlighting the need for further enhancements in both the accuracy and comprehensiveness of gene annotations [2]. Ensuring high genomic stability and coherence is paramount in functional genomics studies, while precise annotated files are indispensable for extracting the vast array of information present within genomic information. The currently utilized 9930 V3 version stands as the most complete reference genome for cucumber, attaining a BUSCO score of 95.4% based on angiosperm embryophyte benchmarks. Yet its gene structure annotations exhibit inadequacies in their comprehensiveness. In this study, we manually reannotated the cucumber gene models by incorporating 64 previously published RNA datasets and 9 newly generated strand-specific RNA datasets, covering various developmental stages (e.g., stem, flower, and fruit tissues) and experimental conditions. The quality of the updated annotation file, which includes 24,145 protein-coding genes, was evaluated using the BUSCO tool, resulting in a completion rate of 96.9%. The nomenclature for the newly added or revised genes followed the format of CsaV3.1_XGXXXXX. Validation of the annotations was performed through the identification of 21 genes using PCR assays and Sanger sequencing. Moreover, the gene expression across diverse tissues was investigated, revealing varying expression patterns, especially those specific to particular tissues. In conclusion, the newly established gene annotation file and its related expression data will serve as a solid and reliable basis for functional studies on cucumber.

## 2. Results

### 2.1. Reannotation of the Cucumber Genome through Manual Operation

To enhance the utility of the V3 reference genome, a thorough manual reannotation was conducted through the integration of publicly accessible RNA-seq data alongside newly acquired strand-specific RNA-seq information. By leveraging 64 RNA-seq libraries derived for the Chinese Long cucumber variety, encompassing diverse tissue types and experimental conditions, a substantial body of data comprising 1.6 billion paired-end reads was generated. Nine strand-specific sequencing data were employed to analyze the gene orientation in root, stem, and leaf tissues, with three replicates for each. Through meticulous examination of the RNA data, sequences showing transcriptional activity were manually curated. This effort led to the creation of an updated annotation file containing 24,145 predicted protein-coding genes (Figure 1).

In contrast to the prior annotation, the updated annotation file reflects alterations or supplements to 7721 genes, constituting 31.99% of the overall gene tally. Genes remaining unaltered preserve their original designations in the revised iteration, while adjusted or newly integrated genes are denoted by the CsaV3.1_XGXXXXX nomenclature. Essential statistical comparisons between pre- and post-modifications have been tabulated in Table 1. The mean exon count per gene witnessed an increment from 5.22 to 5.27, and the average coding sequence (CDS) length elongated from 1189 to 1247 nucleotides. The BUSCO tool is recognized for its reliability in evaluating the completeness of genome annotations, with a higher score indicating a more comprehensive annotation. The updated genome annotation attained a completion rate of 96.9% according to BUSCO, exceeding the previous score of 95.4%. This objectively confirms the enhanced completeness of the new annotation version in comparison to the previous version.

### 2.2. Functional Annotation of Genes in the New Annotation File

The updated annotation file was scrutinized for alterations in the coding sequences. Functional annotation of all the protein sequences was conducted utilizing the GO, KEGG, and iTAK databases. Gene categorization into specific GO terms was accomplished via eggNOG-mapper, yielding an assignment of 46.15% (11,142 transcripts) of the total 24,145 transcripts to distinct GO terms, indicating enhancement from the previous assignment of 45.58% (11,086 transcripts). Furthermore, annotation of protein sequences using the Kobas online tool led to the assignment of 84.15% (20,317 transcripts) to specific KEGG pathways, surpassing the previous assignment of 82.93% (20,167 transcripts).

The iTAK software tool (http://itak.feilab.net/cgi-bin/itak/index.cgi, accessed on 12 April 2024) was employed for the identification of transcription factors (TFs), transcriptional regulators (TRs), and protein kinases (PKs) based on protein or nucleotide sequences, with subsequent classification of individual TFs. The updated annotation revealed a total of 1792 TFs/regulators and 801 protein kinases. Modifications to the gene models of 772 TFs/regulators and 349 protein kinases were made in the revised annotation. Noteworthy alterations were observed in the abundance of specific TF families; for instance, the AP2/ERF family decreased from 144 to 139 genes, the C2H2 zinc finger family decreased from 105 to 101 genes, and the C3H family decreased from 47 to 46 genes, while the C2C2 family increased from 81 to 82 genes. Additionally, structural revisions were implemented for 43 RLK-Pelle_DLSV genes, 20 RLK-Pelle_LRR-III genes, and 16 RLK-Pelle_LRR-XI-1 genes in the protein kinase analysis. In summary, the new annotation file underwent substantial modifications. All the predicted gene functions, as determined using the plaBi database, are provided in Appendix A.

### 2.3. Identification of Novel Gene Features in the New Annotation File

Moreover, apart from reannotating established genes, a comparative assessment was carried out utilizing the gffcompare tool (v0.12.6) to analyze modifications in the annotation datasets. The findings indicated that 16,424 genes retained their original annotations, while 1490 newly identified genes were detected and 6231 original genes were refined. The dispersion of these novel genes on the cucumber chromosomes displayed an uneven distribution, with 853 genes located on Chromosome 2 and 1334 genes on Chromosome 3.

Out of the 7721 newly identified gene structures, 5400 (69.93%) were associated with KEGG terms, and 2716 (35.18%) were linked to GO terms. Using the RNA-seq data and the updated annotation file, the expression profiles of these novel genes were evaluated in roots, stems, and leaves. Among the total genes, 5746 presented an average expression level above 2 transcripts per million (TPM) in the analyzed tissues. Subsequently, a visual representation was constructed to illustrate the top 50 genes with the highest expression levels in each tissue (Figure 2A). The majority of these genes exhibited expression values ranging from 2 to 200 TPM, reflecting distinct tissue-specific expression patterns. Notably, of the top 50 highly expressed genes, 13 were predominantly expressed in the leaves, 21 in the stems, and 16 in the roots. According to the GO enrichment analysis, they are related to pathways such as environmental response and cytoplasmic activities (Figure 2B).

The results reveal new gene characteristics in the cucumber genome, offering insights into their distribution and tissue-specific expression patterns. Additional research is necessary to clarify the specific roles and importance of these recently identified gene features in cucumber biology.

### 2.4. Validation of Selected Novel Transcripts

In order to validate the accuracy and dependability of the revised annotation file, a total of 21 newly annotated genes at the genomic level were randomly selected for verification. The coding sequences (CDS) of these predicted genes were amplified using the primers detailed in Appendix A, followed by confirmation through Sanger sequencing.

The preceding annotation file demonstrated multiple instances of gene annotation inaccuracies, errors, and omissions, as depicted in Figure 3. Initially, there were instances of excessively extended untranslated regions (UTRs), notably exemplified in the initial annotation file for the gene *CasV3_1G000090*. Secondly, numerous genes were annotated with an incorrect exon count, as exemplified by the gene *CsaV3_4G028190*, which exhibited three erroneously annotated exons in the original file. Thirdly, misconceptions in gene annotation were identified, such as the merging of gene entities, as illustrated by *CasV3.1_1G02457*, initially annotated as separate genes, *CasV3_1G00097* and *CasV3_1G00098*. In addition, the absence of partial UTR structures in previous annotations was detected, notably in the gene *CasV3_3G03778*. Furthermore, discrepancies in gene expression profiling led to the exclusion of highly expressed genes, like *CasV3.1_6G01238*. Moreover, instances of missing exons were noticed, such as in the case of *CasV3.1_7G00744*, which lacked a terminal exon. Lastly, discrepancies in gene length were observed compared to the RNA data, particularly evident in the shortened length of the gene *CsaV3_7G00815* at one end according to the RNA data.

Through the implementation of validation procedures, our objective is to verify the accuracy of the updated annotation document and rectify any flaws, discrepancies, or oversights detected in the prior annotation. These results enhance the knowledge of the cucumber genome, particularly its genetic architecture, in a more thorough and reliable manner.

### 2.5. Expression Profiles of Genes in the Novel Annotation File Based on RNA-Seq Data

Highlighting expression profiling is an effective tool for gene recognition. The study calibrated the expression matrix encompassing roots, stems, and leaves derived from the “9930” cucumber inbred line using TPM. Among these plant parts, this research observed 22,190 genes exhibiting significant expression, with a mean TPM value exceeding 1 in 17,575 genes. Each sampled portion contained an approximation of 19,466 to 20,326 expressed genes, with 19,842 genes consistently expressed across all three types of tissues (Appendix A).

The study categorized the expressed genes (22,190 in total) into six distinct clusters considering their unique expression tendencies across various tissues (Figure 4). The groups comprised 3553, 5444, 4240, 2292, 3969, and 2978 genes, respectively. These results reinforce the inherent tissue specificity of gene expression. Interestingly, groups 1 and 5 manifested marked expression in the stem tissues, while groups 3 and 4 showcased escalated expression in the leaf tissues. Predominantly, roots were the major sites of expression for groups 2 and 6.

For an in-depth understanding of the functional role of the grouped genes, exhaustive annotations were carried out (Appendix A). Group 3 and group 4 genes—highly expressed in the leaves—showed a rich presence in metabolic pathways linked to plant hormone signaling, photosynthesis, and metabolism (Appendix A). Meanwhile, the genes from group 6, mainly found expressed in the roots, indicated an enrichment in energy-related metabolic pathways, including polysaccharide and lipid metabolism (Appendix A).

A tissue specificity index analysis was applied to delve into tissue-specific genes in the roots, stems, and leaves using a Perl script. Genes with TAU values exceeding 0.5 were deemed to show tissue-specific expression. In total, 11,868 genes exhibited unique tissue specificity. Such tissue-specific genes were consistently distributed over all seven chromosomes (Figure 5A), and their expression levels were illustrated using a heatmap (Figure 5B). In light of their high expression, functional annotations of genes using Gene Ontology (GO) and Kyoto Encyclopedia of Genes and Genomes (KEGG) analyses elucidated the tissue-specific functions. For instance, of the 4540 genes identified as being specifically expressed in the stems, functional annotation highlighted their role in plant hormone signaling and genetic signal transduction.

### 2.6. Transcripts with Diverse Expression Levels across Different Tissues

The analysis of a total of 17,575 genes revealed an impressive subset of 14,185 genes associated with KEGG information, having consistently shown an average TPM value greater than 1 across all three tissue types. These annotated genes were primarily engaged in crucial metabolic pathways, which include polysaccharide metabolism, lipid metabolism, plant hormone signaling, photosynthesis, and broader cellular metabolism. Interestingly, within these annotated genes, a noteworthy group of 237 were identified as transcription factors, and 148 were categorized as protein kinases. Some unique transcription factors and protein kinases demonstrated remarkably similar expression patterns, indicating near-identical expression levels in the roots, stems, and leaves (Figure 6).

## 3. Discussion

### 3.1. Enhancing Functional Genomics Research into Cucumber through Accurate Gene Annotation

Cucumber, a major global vegetable, exhibits unique genetic characteristics, such as a rather compact genome size (approximately 260 Mb) and few chromosomes (2*n* = 14). Prior research has emphasized the potency of top-tier genomic resources in pinpointing crucial Quantitative Trait Loci (QTLs) and potential genes related to agronomic traits [15,16,17,18,19]. Notably, fluctuations in the expression of five particular genes (*CsaV3_1G028310*, *CsaV3_2G006960*, *CsaV3_3G009560*, *CsaV3_5G031320*, and *CsaV3_6G031260*) are associated with stem thickness in cucumber [20]. Also, a specific 1449 bp sequence insert within CsMLO8’s coding section at the pm-s5.1 location notably bolsters powdery mildew resistance in cucumber stems [21]. Past phenotypic research stretching across the previous century has elucidated that the fruit neck length in cucumber is predominantly governed by additive genetic components rather than environmental influences. This revelation, in conjunction with the identification of the *HECATE1* gene in cucumber (*CsHEC1*) and its pronounced correlation with fruit neck length, has greatly augmented our insight into the regulatory controls over the variation in fruit neck length in cucumber [22,23]. In essence, high-grade assembled genomic datasets function as invaluable tools for examining crop characteristics and gene operations, thereby galvanizing progress in both scientific investigations and industrial advancements within the vegetable sector.

The criticality of accurate gene model annotations for efficiently utilizing genomic information, enhancing the analysis precision of extensive biological datasets, and forwarding molecular biology experimentations cannot be overstated [24]. Previously, gene structures were predominantly annotated according to computational predictions, de novo prediction, homology-based prediction, and RNA-seq methodology included. However, it is important to note that these techniques often generate annotation files replete with numerous miscalculations, resulting from the inherent restrictions of the utilized algorithms. Historically, extensive algorithmic resources were essential for alignment and assembly tasks. Insufficient algorithmic precision in meeting intricate annotation demands inevitably results in alignment inaccuracies and reduced assembly precision. Particularly in species with intricate genomes, the error rates tend to be elevated, leading to frequent challenges in region annotation. To offset these anomalies, researchers have found employing an iterative reannotation process using varied methodologies and datasets beneficial for obtaining more expansive genomic and transcriptomic data. Encouragingly, these tactics have shown promising results in model plants and an array of economically valuable crops, such as Arabidopsis, orange, rice, maize, peach, and strawberry [4,5,6,7,8,25].

As well as automated annotation grounded in algorithms, manual adjustment via the visualization of RNA data stands as a notably pragmatic and efficacious mode for attaining precise annotation files. A case in point is the research on the Red5 kiwifruit genome, in which gene structures were meticulously modified employing WebApollo software [9]. Moreover, GSAman software has triumphed in annotating the peach genome and decoding the genetics of sweet potatoes [8,10]. This strategic method for annotation significantly boosts the accuracy of gene models. On the downside, the task of manipulating gene structures at the genomic level presents a strenuous endeavor, with the use of manual annotation software adding to the complexity in an appreciable way.

Nevertheless, it is widely recognized that the manual refinement of gene models enables the most precise predictions about gene behavior. Our study has led to the emergence of updated genome annotation, meticulously tailored and drawing upon 64 open-source RNA-seq datasets and 9 stranded-specific RNA-seq sets collected from root, stem, and leaf samples. Meticulous alignment of these collections with the cucumber 9930_V3 reference genome led to the calibration of 7721 earlier non-annotated gene structures. For validation, Sanger sequencing was conducted on 21 distinct genes, affirming their perfect congruence with the anticipated gene results. The outcomes of this study equip other scientists in similar domains with a potent and enhanced gene structure annotation file, driving forward innovation in the bioinformatics and molecular biological investigations of cucumber.

### 3.2. Gene Expression Profiles Provide Support for Gene Identification

Transcriptomic expression analysis serves as a fundamental tool for pinpointing gene candidates associated with various factors. A plethora of studies on fruit quality have underscored its significance by utilizing this method to discern primary gene candidates contributing to cucumber’s resistance to powdery mildew [26]. Additionally, through the employment of expression analysis, an array of genes contributing to the buildup of compounds such as anthocyanins, carotenoids, terpenes, acids, and flavonoids across different fruit organs have been accurately established [27,28,29,30,31,32,33]. Offering a wealth of information on the gene expression levels across varied tissues and conditions, transcriptomic expression analysis vitally assists targeted screening for specific candidate genes. 

This study annotated an impressive number of 24,145 genes, and 7721 of these genes experienced modifications or additions. This work carried out an RNA-seq expression analysis of roots, stems, and leaves and found that over 70%—precisely 17,575 transcripts—showed expression levels above 1 TPM. Interestingly, around 16% of genes, accounting for 3957, sustained their gene structure models without changes, despite having minimal expression. It is worth mentioning that the Version 1.0 genome’s gene structures were computationally anticipated via de novo and homology-based approaches. This indicates these genes may stand unexpressed or their expression may occur under conditions not investigated in our study—highlighting the likely impact of including RNA-seq data from various tissues and experimental setups [25]. Analyzing RNA expression to reveal tissue-specific expression trends creates fundamental groundwork for handpicking appropriate genes contributing to particular traits. For instance, by using such tissue-specific expression layouts, several genes linked to wilt disease resistance have been detected in watermelon [34]. Similarly, a study into the tissue-specific spatiotemporal expression patterns in lotus disclosed genes associated with phenolic compounds [35]. Further, analysis of tissue-specific expression facilitated the pinpointing of genes associated with head formation in Chinese cabbage [36]. The findings from these studies highlight the potential role of genes expressed uniquely in different tissues in various metabolic pathways within those specific tissues.

## 4. Materials and Methods

### 4.1. Construction and Sequencing of RNA Libraries

To update the annotation of the cucumber 9930_V3 reference genome, 64 RNA datasets from diverse tissues and experimental conditions were employed, sourced from the National Center for Biotechnology Information (NCBI) website (Appendix A). Moreover, new strand-specific RNA sequencing was conducted on root, stem, and leaf tissues, with three replicates per tissue, to document the transcript orientation.

### 4.2. Reannotation of the 9930_V3 Reference Genome

In order to enhance gene structure annotation and overcome algorithmic limitations, manual annotation based on RNA-seq data was employed. Initially, a genome index was created using hisat2-Build, and clean reads were aligned with the cucumber 9930_V3 reference genome using Hisat2. The resulting alignments were then converted into BAM files using SAMtools (v1.13) [37]. Subsequently, duplicate reads were eliminated using SAMtools’ markdup function, and the BAM files were sorted and indexed. The sorted BAM files were then utilized for gene structure adjustment via GSAman (v.0.8.2, available online: https://tbtools.cowtransfer.com/S/a11146181df14f (accessed on 8 October 2023)). The structures of all the expressed genes underwent visual confirmation through inspection of the BAM files.

### 4.3. Validation of Novel Transcripts

Total RNA was isolated from the foliage of the Chinese North variety of the cultivated cucumber inbred line 9930 using the accuracy^®^ Universal Plant RNA Extraction Kit (ACCURATE BIOTECHNOLOGY(HUMAN) Co., Ltd., Changsha, China). Subsequently, the isolated RNA was transcribed in the opposite direction to generate complementary DNA (cDNA) through the use of the Evo M-MLV Plus 1st Strand cDNA Synthesis Kit. The coding regions (CDS) of 21 genes selected at random were amplified using the specific primers detailed in Appendix A and the accuracy High-Fidelity DNA polymerase. The resulting amplified segments underwent Sanger sequencing, followed by alignment with the novel annotation sequences for verification.

### 4.4. Functional Annotation of Genes in the Novel Annotation File

Functional annotation of genes in the new annotation file was conducted through the utilization of the eggNOG-mapper (v2.1.12), Kobas (v.3.0), iTAK Online (v.1.6), and Mercator (v.3.6) platforms [38,39,40,41]. Protein sequences underwent analysis on the eggNOG-mapper and Kobas websites using the default parameters. Alignment of the CDS sequences with databases such as TAIR10, SwissProt/UniProt plant proteins, Augustus Models (JGI Chlamy Release 4), TIGR 5 rice proteins, and the NCBI conserved domain database was performed using Mercator (v.3.6).

### 4.5. Gene Expression Profiles in the Novel Annotation File

RNA-seq data obtained from three tissue specimens underwent alignment with the 9930_V3 reference genome and normalization employing transcripts per million (TPM) values, which were informed by the updated gene structure annotation. Utilization of the Mfuzz package in the R programming language facilitated the categorization of the expressed genes into six distinct clusters. Subsequently, Perl scripts were employed to ascertain the tissue specificity coefficient for individual genes [42], with genes exhibiting a coefficient surpassing the threshold of 0.5 being selected for further analysis.

## 5. Conclusions

The unveiling of the high-quality 9930_V3 cucumber genome in 2019 denoted a significant landmark. To elevate the precision of the gene prediction model, comprehensive refinements of the genome structure annotation file were undertaken. This included detailed reannotation of the cucumber 9930_V3 reference genome through a robust dataset composed of 64 publicly accessed RNA-seq datasets and 9 strand-specific RNA-seq datasets. After stringent validation via GO, KEGG, BUSCO, and RT-PCR analyses, the enhanced superiority of the newly annotated file to its predecessor was unequivocally established. It is worth mentioning that this research facilitated the amendment and incorporation of 7721 gene structures into the improved annotation file. By taking advantage of this augmented annotation, the research performed a deeper investigation into the expression profiles of genes in the tissues of the roots, stems, and leaves. This enhanced version of the cucumber 9930_V3 genome annotation gives researchers in bioinformatics and molecular biology an extraordinarily accurate gene structure prediction model, paving the way for more profound insights into cucumber biology.

## Figures and Tables

**Figure 1 plants-13-01604-f001:**
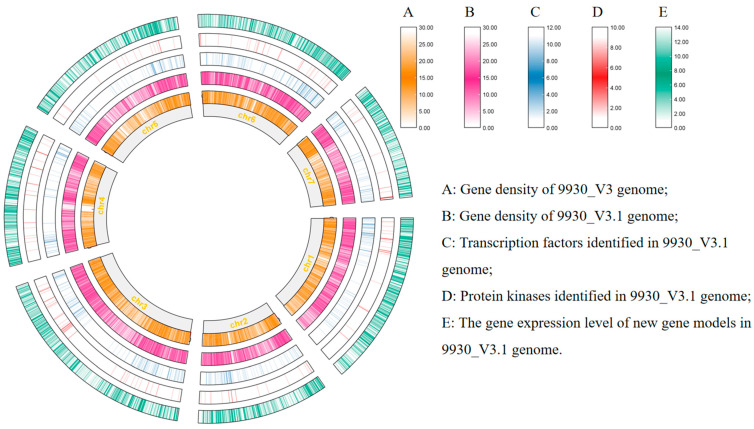
Characterization of the updated gene annotation of the 9930 cucumber genome. (**A**) Gene density in the 9930_V3 reference genome; (**B**) Gene density in the updated 9930_V3.1 reference genome; (**C**) Transcription factors identified in the 9930_V3.1 reference genome; (**D**) Protein kinases discovered in the 9930_V3.1 reference genome; (**E**) Gene expression levels of the newly annotated gene models in the 9930_V3.1 reference genome. The image was generated by TBtools [14] with default parameters.

**Figure 2 plants-13-01604-f002:**
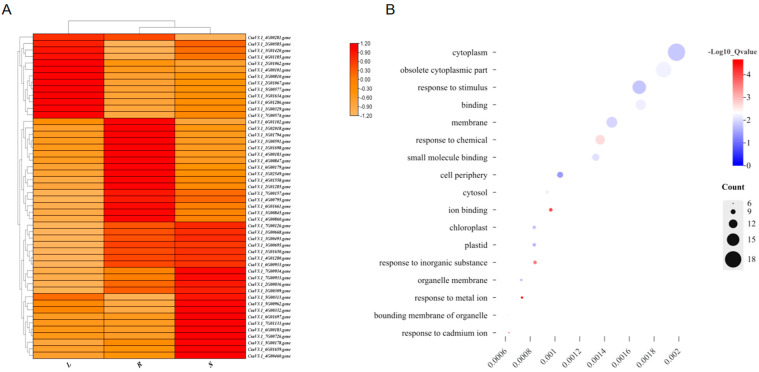
Heatmap (**A**) illustrating the top 50 novel genes with the highest relative expression levels among the 7721 novel discovered genes and their GO enrichment (**B**). (**A**) was generated using TBtools [14] with default parameters, and (**B**) was made with ggplot2 package in R (v4.3.3).

**Figure 3 plants-13-01604-f003:**
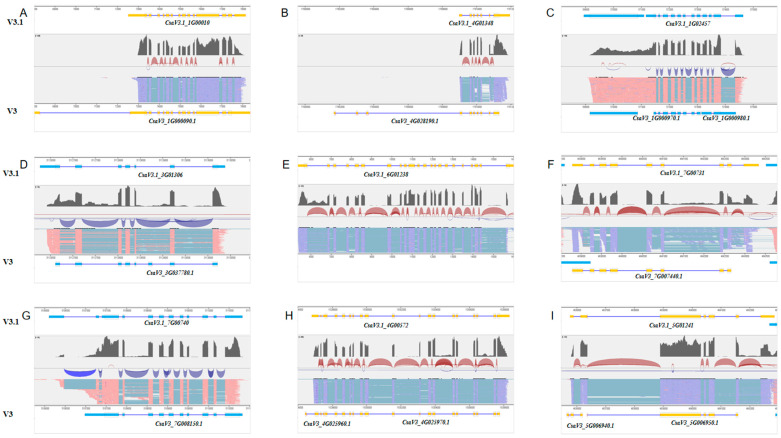
Gene adjustments and new additions to the updated annotation based on RNA-seq mapping results. (**A**) Premature termination of the *CasV3_1G000090* gene prediction. (**B**) Reduction of three exons in the *CsaV3_4G028190* gene. (**C**) Fusion of *CsaV3_1G00097* and *CsaV3_1G00098* into a single gene. (**D**). Delayed termination of the *CsaV3_3G03778* gene prediction. (**E**). Addition of the *CsaV3.1_6G01238* gene. (**F**). Delayed termination and the inclusion of an additional intron in the predicted *CsaV3_7G00744* gene. (**G**). Delayed termination and the inclusion of two additional introns in the predicted *CsaV3_7G00815* gene. (**H**). Fusion of *CsaV3_4G02596* and *CsaV3_4G02597* into a single gene with a delayed termination. (**I**) Mutual impact of gene expression results among *CsaV3_5G00694*, *CsaV3_5G00695*, and *CsaV3_5G00696*. The images were output from GSAman (v.0.8.2).

**Figure 4 plants-13-01604-f004:**
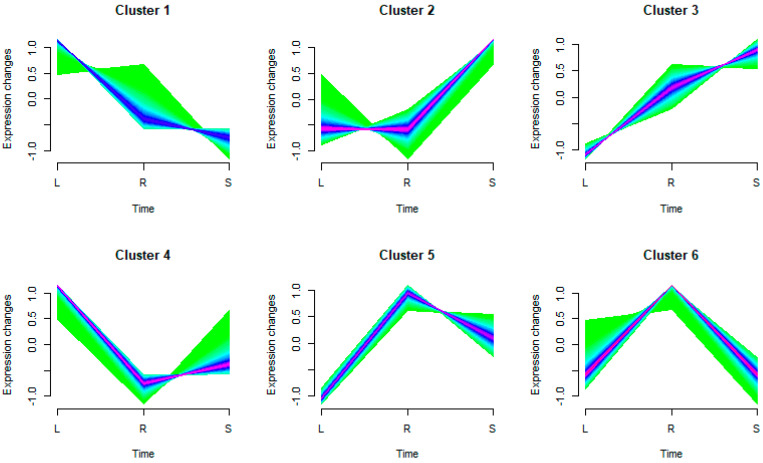
Genes were clustered according to their expression patterns in the transcriptome data from three tissue samples. The shades of color represent the density of genes, the darker the color, the more genes are concentrated. The image was generated using TBtools [14] with default parameters.

**Figure 5 plants-13-01604-f005:**
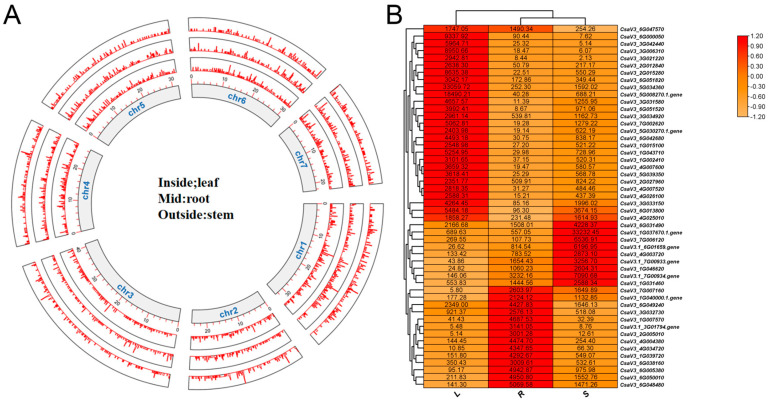
Tissue-specific genes located across the seven chromosomes (**A**), and heatmap showing the expression levels of genes expressed in specific tissues: leaf, root, and stem (**B**). The images were generated using TBtools [14] with default parameters.

**Figure 6 plants-13-01604-f006:**
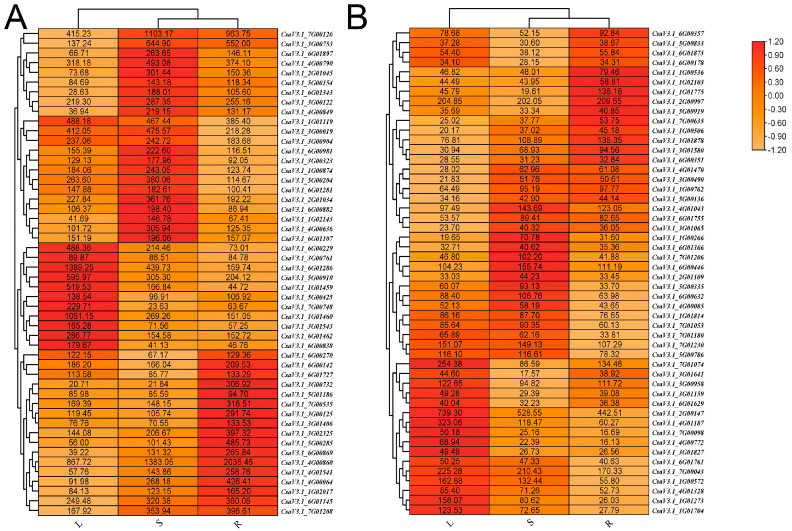
Heatmaps depicting the expression levels of transcription factor (**A**) and protein kinase (**B**) transcripts across three tissues. The images were generated using TBtools [14] with default parameters.

**Table 1 plants-13-01604-t001:** Summary of the new cucumber genome annotation.

	V3	V3.1
Number of genes	24,317	24,145
Mean mRNA length (bp)	1716	1819
Mean exon number	5.22	5.27
Mean CDS length (bp)	1128	1131
Genes with GO terms	11,086	11,142
Transcripts with KEGG terms	20,167	20,317
Complete BUSCOs	95.4%	96.9%
Complete and single-copy BUSCOs	91.3%	92.6%
Complete and duplicated BUSCOs	4.1%	4.3%
Fragmented BUSCOs	1.5%	0.7%
Missing BUSCOs	3.1%	2.2%

## Data Availability

The raw reads of nine strand-specific RNA-seq data were submitted to the National Genomics Data Center under the accession number PRJCA025018. The updated version of the annotation in gff3 file format can be found in the uploaded Appendix A.

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
