# Peer review of "Updated Gene Prediction of the Cucumber (9930) Genome through Manual Annotation"

_plants, 2024, doi:10.3390/plants13121604_

Round 1

Reviewer 1 Report

Comments and Suggestions for Authors

1. In this work, the authors reported an improved annotation for the cucumber genome 9930v3.0. Over one third of the genes were re-annotated. Such work in meaningful and useful for the community if properly documented and accessible, which, however, is not enough at current version.

2. In the title, make sure to specify this improved annotation is for the ‘9930’ cucumber.Since the current version is v3.0, it may be appropriate to naming your new version 9930v3.1 as showing in the gene ID. Using v1.0 and v2.0 will confusing the readers and potential users.

3. The early annotation may used machine annotation plus experimental data such as RNA-Seq, homology search. The current 'manual' annotation relied primarily on RNA-Seq data. There are other resources that be helpful to improve the annotation, which the authors may wish to explore. I will be nice to examine additional available. Plus, it will very helpful to indicate/summarize main reasons or causes for the incorrect annotation in 9930v3.0. Such work may be useful beyond cucumber.

4. it is critical to make the work useful for the community. Thus, a complete list of all annotated genes in 9930v3.1 side by side with 9930v3.0  is needed. Indicate the genome locations of all genes . For example, a gff3 file indicating all components of a gene annotation and the position of each feature in the 9930v3.0 genome.

5. As far as I remember, several published papers discussed potential mis-annotation of some genes in 9930v3.0. Those could also be used to validate the correction of the new version.

6. Table S7: Please make it human readable.

Comments on the Quality of English Language

Largely OK although additional language editing is nice to improve readability.

Author Response

In this work, the authors reported an improved annotation for the cucumber genome 9930v3.0. Over one third of the genes were re-annotated. Such work in meaningful and useful for the community if properly documented and accessible, which, however, is not enough at current version.

1. Comment: In the title, make sure to specify this improved annotation is for the ‘9930’ cucumber.Since the current version is v3.0, it may be appropriate to naming your new version 9930v3.1 as showing in the gene ID. Using v1.0 and v2.0 will confusing the readers and potential users.

Response: Thank your for your comments. We have specified the cucumber genome in the tile: “Updated gene prediction of cucumber (9930) genome through manual annotation”. To avoid potential confusion, ‘version 1.0’ (‘v1.0’) and ‘version 2.0’ (‘v2.0’) have been replaced with ‘V3.0’ and ‘V3.1’, respectively.

2. Comment: The early annotation may used machine annotation plus experimental data such as RNA-Seq, homology search. The current 'manual' annotation relied primarily on RNA-Seq data. There are other resources that be helpful to improve the annotation, which the authors may wish to explore. I will be nice to examine additional available. Plus, it will very helpful to indicate/summarize main reasons or causes for the incorrect annotation in 9930v3.0. Such work may be useful beyond cucumber.

Response: Despite annotation methods like machine annotation using database-based or experimental data have been widely used and made great progress, the accuracy of these algorithms is still far from perfect, therefore, for downstream research, the annotation file should be manually "polished", which has been done in several crop species, e.g. Arabidopsis, orange, rice, maize, peach and strawberry[4–8,30]. To better interpret this, we have revised this part in the discussion.

3. Comment: it is critical to make the work useful for the community. Thus, a complete list of all annotated genes in 9930v3.1 side by side with 9930v3.0 is needed. Indicate the genome locations of all genes . For example, a gff3 file indicating all components of a gene annotation and the position of each feature in the 9930v3.0 genome.

Response: The list of all annotated genes in 9930v3.1 side by side with 9930v3.0 was listed in an additional supplementary table (Table S8-2). The gff3 file was also provided in the supplementary file.

4. Comment:As far as I remember, several published papers discussed potential mis-annotation of some genes in 9930v3.0. Those could also be used to validate the correction of the new version.

Response: Yes, we would very much like to validate these potential miss-annotated genes with our manually annotated gene prediction, however, honestly speaking, we have not been able to find these published potential mis-annotation genes. It would be appreciated if you could provide information on these miss-annotated genes. We will also keep an eye on these articles and validate these genes in the future.

5. Comment: Table S7: Please make it human readable.

Response: Table S7 has been revised for better understanding.

Reviewer 2 Report

Comments and Suggestions for Authors

The study focuses on enhancing the accuracy of predicted gene models in the cucumber genome by incorporating a comprehensive dataset comprising 64 published RNA-seq data and 9 new strand-specific RNA-seq datasets from diverse tissues. This manual comparison led to the development of an updated annotation file, designated as version 2.0. The cucumber 9930 variety Chinese Long serves as the reference for this study.

The updated annotation file (version 2.0) encompasses a notable augmentation, featuring 24,145 predicted genes, a substantial increase from the previous version's count of 24,317 genes. The revised annotation demonstrates a higher BUSCO value of 96.9%, underscoring the improved accuracy and completeness of the gene prediction model. Notably, 6,231 transcripts were adjusted, and 1,490 transcripts were newly added, collectively constituting 31.99% of the overall gene tally. Newly added and adjusted genes are systematically renamed (CsaV3.1_XGXXXXX), while retaining the original designations for unaltered genes. To validate the modifications, a random selection of 21 modified/added genes underwent rigorous RT-PCR analyses, ensuring the credibility of the updated annotations. Leveraging newly obtained transcriptome data, the study scrutinized tissue-specific patterns of gene expression, aligning them with the revised gene prediction model.

In conclusion, the updated annotation file (version 2.0) represents a significant advancement in elucidating the gene structure landscape of the cucumber genome. By amalgamating extensive RNA-seq datasets and employing meticulous manual comparison techniques, the study has substantially enriched the gene prediction model, thereby furnishing researchers with invaluable resources for advancing genetic studies in cucumber.

The manuscript requires major revisions, particularly regarding the clarity of figures and detailed information provided in figure legends. Figures 2 through 7 require improvement in size and clarity. Additionally, inclusion of GO and KEGG data for the newly annotated genes would enhance the manuscript's completeness.

- Check thoroughly for format issues, such as spacing around citations.

- Clarify references to the cucumber variety Chinese Long.

- Move paragraphs L72-77 to the end of the introduction for better organization.

- Provide detailed information for figure legends, including how the heatmap in Figure 2 was generated (tools, parameters, etc.).

- Increase the size of the heatmap in Figure 2 for better visibility.

- Enlarge images in Figures 3, 4, 5, 6, and 7 to improve visibility.

- Simplify references to the company (AC- 345 CURATE BIOTECHNOLOGY(HUMAN) CO.,LTD, ChangSha, China) to just the company name after the first description in the manuscript.

After addressing these issues and revisions, the manuscript will be suitable for publication.

Author Response

The study focuses on enhancing the accuracy of predicted gene models in the cucumber genome by incorporating a comprehensive dataset comprising 64 published RNA-seq data and 9 new strand-specific RNA-seq datasets from diverse tissues. This manual comparison led to the development of an updated annotation file, designated as version 2.0. The cucumber 9930 variety Chinese Long serves as the reference for this study.

The updated annotation file (version 2.0) encompasses a notable augmentation, featuring 24,145 predicted genes, a substantial increase from the previous version's count of 24,317 genes. The revised annotation demonstrates a higher BUSCO value of 96.9%, underscoring the improved accuracy and completeness of the gene prediction model. Notably, 6,231 transcripts were adjusted, and 1,490 transcripts were newly added, collectively constituting 31.99% of the overall gene tally. Newly added and adjusted genes are systematically renamed (CsaV3.1_XGXXXXX), while retaining the original designations for unaltered genes. To validate the modifications, a random selection of 21 modified/added genes underwent rigorous RT-PCR analyses, ensuring the credibility of the updated annotations. Leveraging newly obtained transcriptome data, the study scrutinized tissue-specific patterns of gene expression, aligning them with the revised gene prediction model.

In conclusion, the updated annotation file (version 2.0) represents a significant advancement in elucidating the gene structure landscape of the cucumber genome. By amalgamating extensive RNA-seq datasets and employing meticulous manual comparison techniques, the study has substantially enriched the gene prediction model, thereby furnishing researchers with invaluable resources for advancing genetic studies in cucumber.

The manuscript requires major revisions, particularly regarding the clarity of figures and detailed information provided in figure legends. Figures 2 through 7 require improvement in size and clarity. Additionally, inclusion of GO and KEGG data for the newly annotated genes would enhance the manuscript's completeness.

Comment:

- Check thoroughly for format issues, such as spacing around citations.

Response: The format around citations has been corrected accordingly.

- Clarify references to the cucumber variety Chinese Long.

Response: We have specified the cucumber genome in the tile: “Updated gene prediction of cucumber (9930) genome through manual annotation”.

- Move paragraphs L72-77 to the end of the introduction for better organization.

Response: Lines 72-77 have been moved to the end of the introduction as suggested.

- Provide detailed information for figure legends, including how the heatmap in Figure 2 was generated (tools, parameters, etc.).

Response: The images were all generated by Tbtools with default parameters. We have added this in each figure legend and the material and methods section.

- Increase the size of the heatmap in Figure 2 for better visibility.

Response: Figure 2 has been enlarged to increase visibility.

- Enlarge images in Figures 3, 4, 5, 6, and 7 to improve visibility.

Response: Figures 3, 4, 5, 6, and 7 have been enlarged to increase visibility.

- Simplify references to the company (AC- 345 CURATE BIOTECHNOLOGY(HUMAN) CO.,LTD, ChangSha, China) to just the company name after the first description in the manuscript.

Response: The name of the company has been simplified as suggested.

Round 2

Reviewer 1 Report

Comments and Suggestions for Authors

I have the following suggestions for additional minor revision:

1. In Table S8-1; add one more column to include annotated function(s) for each gene. Such info seems to be in Table S7 but less explicit. This is important for regular users who do not have much idea about gene annotation.

2. Gene ID: I Table S8-2, newly annotated genes and 9930v3.0 were assigned somewhat different IDs. I suggest consistency in naming the genes. 1) Use the same prefix (V3.1) for all genes regardless old and new genes; 2) Use the same number of digits; For example

CsaV3.1_1G00037  -->  CsaV3.1_1G000037.1
CsaV3_1G000290.1 --> CsaV3.1_1G000290.1

Author Response

I have the following suggestions for additional minor revision:

Comment:1. In Table S8-1; add one more column to include annotated function(s) for each gene. Such info seems to be in Table S7 but less explicit. This is important for regular users who do not have much idea about gene annotation.

Response: Thank you for taking the time to review our revised manuscript and providing your expert opinion. As you mentioned, we need to consider the perspective of readers and cater to the needs of the majority, such as those who may not have a deep understanding of gene annotation. This is a good point that deserves further consideration. However, after thorough discussion, we found that there are many sources of functional annotation for coding genes, such as GO, KEGG, or the data from the PlaBI database that you mentioned, among others. These are sources of functional annotation for the structural annotation of coding genes, and this information has been presented in Tables S3-S7. Listing these complex and diverse pieces of information in a single table would make it overly cluttered and difficult to read. Therefore, we have separated this various functional annotation information into multiple tables to make it easier for researchers to use. Of course, if you are interested in a particular gene, you can refer to the aforementioned tables to achieve your goal of understanding that gene.

Comment:2. Gene ID: I Table S8-2, newly annotated genes and 9930v3.0 were assigned somewhat different IDs. I suggest consistency in naming the genes. 1) Use the same prefix (V3.1) for all genes regardless old and new genes; 2) Use the same number of digits; For example ï¼š

CsaV3.1_1G00037  -->  CsaV3.1_1G000037.1

CsaV3_1G000290.1 --> CsaV3.1_1G000290.1

Response: Thank you for bringing up the issue regarding gene nomenclature. However, the last version of cucumber annotation was released about five years ago, and since then researchers have used and are currently using this annotation file for functional validation and other studies. Our work aims to make necessary corrections to this widely used version, and there are still many genes that we have not modified. To facilitate researchers studying genes that we have not modified, we have intentionally differentiated gene IDs: modified genes are given new IDs, while the original IDs are retained for genes that have not been modified. Although we understand the reviewers' careful consideration of gene ID assignment, we believe that differentiation in IDs is a better solution to prevent unnecessary confusion.

Reviewer 2 Report

Comments and Suggestions for Authors

Authors diligently carried out revisions according to reviewers' feedback. Now, the manuscript is suitable for publication in its current form. Congratulations on excellent work!

Author Response

Comment: Authors diligently carried out revisions according to reviewers' feedback. Now, the manuscript is suitable for publication in its current form. Congratulations on excellent work!

Response: Thank you for taking the time to review our manuscript and for your feedback on our research. I wish you all the best in your research.